

# Tea saponin reduces the damage of *Ectropis obliqua* to tea crops, and exerts reduced effects on the spiders *Ebrechtella tricuspidata* and *Evarcha albaria* compared to chemical insecticides

Chi Zeng[*], Lingbing Wu[*], Yao Zhao, Yueli Yun and Yu Peng

Hubei Collaborative Innovation Center for Green Transformation of Bio-Resources, College of Life Science, Hubei University, Wuhan, Hubei, People's Republic of China

[*] These authors contributed equally to this work.

Corresponding author
Yu Peng, pengyu@hubu.edu.cn

## ABSTRACT

**Background.** Tea is one of the most economically important crops in China. However, the tea geometrid (*Ectropis obliqua*), a serious leaf-feeding pest, causes significant damage to tea crops and reduces tea yield and quality. Spiders are the most dominant predatory enemies in the tea plantation ecosystem, which makes them potentially useful biological control agents of *E. obliqua*. These highlight the need for alternative pest control measures. Our previous studies have shown that tea saponin (TS) exerts insecticidal activity against lepidopteran pests. Here, we investigate whether TS represents a potentially new alternative insecticide with no harm to spiders.

**Methods.** We investigated laboratory bioactivities and the field control properties of TS solution against *E. obliqua*. (i) A leaf-dip bioassay was used to evaluate the toxicity of TS to 3rd-instar *E. obliqua* larvae and effects of TS on the activities of enzymes glutathione-S-transferase (GST), acetylcholinesterase (AChE), carboxylesterase (CES) and peroxidase (POD) of 3rd-instar *E. obliqua* larvae in the laboratory. (ii) Topical application was used to measure the toxicity of 30% TS (w/v) and two chemical insecticides (10% bifenthrin EC and 50% diafenthiuron SC) to two species of spider, *Ebrechtella tricuspidata* and *Evarcha albaria*. (iii) Field trials were used to investigate the controlling efficacy of 30% TS against *E. obliqua* larvae and to classify the effect of TS to spiders in the tea plantation.

**Results.** The toxicity of TS to 3rd-instar *E. obliqua* larvae occurred in a dose-dependent manner and the $LC_{50}$ was 164.32 mg/mL. Activities of the detoxifying-related enzymes, GST and POD, increased in 3rd-instar *E. obliqua* larvae, whereas AChE and CES were inhibited with time by treatment with TS. Mortalities of *E. tricuspidata* and *E. albaria* after 48 h with 30% TS treatment (16.67% and 20%, respectively) were significantly lower than those with 10% bifenthrin EC (80% and 73.33%, respectively) and 50% diafenthiuron EC (43.33% and 36.67%, respectively). The highest controlling efficacy of 30% TS was 77.02% at 5 d after treatment, which showed no difference to 10% bifenthrin EC or 50% diafenthiuron SC. 30% TS was placed in the class N (harmless or slightly harmful) of IOBC (International Organization of Biological Control) categories for natural enemies, namely spiders.

**Conclusions**. Our results indicate that TS is a botanical insecticide that has a good controlling efficacy in *E. obliqua* larvae, which suggests it has promise as application in the integrated pest management (IPM) envisaged for tea crops.

# INTRODUCTION

Tea, *Camellia sinensis* Kuntze (Theales: Theaceae), is one of the most economically important crops in China, cultivated in vast areas spreading from 37°N–18°S and 122°E–97°W, totaling more than 20 provinces across tropical, subtropical and temperate regions (*Ye et al., 2014*). The tea geometrid, *Ectropis obliqua* (Lepidoptera: Geometridae), is a major pest throughout tea plantations in China (*Zhang et al., 2014*). Its larvae exclusively feed on tea leaves and tender buds, causing severe deterioration in yield and quality (*Ma et al., 2016*). The therapeutic approach of killing this pest with chemicals has been the prevailing control strategy (*Hazarika, Puzari & Wahab, 2001*; *Ehi-Eromosele, Nwinyi & Ajani, 2013*; *Xin et al., 2016*). However, indiscriminate use of chemicals in tea gardens has given rise to a large number of problems, including resurgence of primary pests (*Harmatha et al., 1987*), resistance development (*Gurusubramanian et al., 2008*), undesirable residues in tea products (*Feng et al., 2015*) and environmental contamination (*Saha & Mukhopadhyay, 2013*; *Ye et al., 2014*).

Botanical insecticides are superior to traditional chemical pesticides in terms of several eco-toxicological indicators, i.e., low toxicity to human, rapid degradation, reduced environmental impact (*Bourguet, Genissel & Raymond, 2000*; *Isman, 2006*; *Chermenskaya et al., 2010*) as well as multiple bioactivities. Through deterring oviposition and feeding, regulating growth and being toxic to larvae and adults, they represent an alternative for pest control (*Isman, 2006*; *Chermenskaya et al., 2010*; *Martínez et al., 2015*). These advantages indicate that botanical insecticides can be an ideal option for managing pests in an eco-friendly and economical way (*Abou-Fakhr, Zournajian & Talhouk, 2001*; *Isman, 2006*; *Roy, Mukhopadhyay & Gurusubramanian, 2010*; *Martínez et al., 2015*).

Tea saponin (TS) is extracted from the seeds of plant species belonging to the genus *Camellia* of the family Theaceae, and it can enhance efficiency and solubilization of pesticide as a wetting powder pesticide (*Chen, Zhang & Yang, 2012*). TS has been widely used in pesticides as the main component of environmentally-friendly pesticide additives (*De Geyter et al., 2007*). *Chaieb (2010)* concluded that the insecticidal activity of saponins is due to a disturbance of the synthesis of ecdysteroids, protease inhibitors or cytotoxic to certain insects. TS, which has a strong insecticidal activity against a broad range of insect types and stages, has potential to be used as a natural insecticide (*Potter et al., 2010*; *Cai et al., 2016*). In our previous studies, we found that 30% TS (w/v; 300 mg/mL) exerted a strong toxic effect on *E. obliqua* larvae (*Hao et al., 2011*; *Liu et al., 2012*).

Biological control has gained recognition as an essential component of successful integrated pest management (IPM) (*Murphy & Briscoe, 1999*; *Jacobsen, Zidack & Larson, 2004*;

*Yang et al., 2017*). Predatory natural enemies play key functional roles in the biological control of IPM (*Rutledge, Fox & Landis, 2004*), and spiders are the most dominant predatory natural enemies in the tea plantation ecosystem (*Chen et al., 2004*; *Das, Roy & Mukhopadhyay, 2010*). *Hu et al. (1994)* used feeding trials in the laboratory to show that *Ebrechtella tricuspidata* and *Evarcha albaria* prey on the larvae of *E. obliqua*. *Yang et al. (2017)* have demonstrated that these two species exhibit maximum potential in taking control of *E. obliqua* larvae, which will be disturbed if chemicals with adverse effects are used. Allowing for that, it has become necessary to find alternative, as well as effective biodegradable insecticides with greater acceptability than chemical insecticides (*Roy, Mukhopadhyay & Gurusubramanian, 2010*).

The physiological and metabolic functions of insects have frequently been reported to be influenced by chemicals or host plant variety (*Cai et al., 2016*). Likewise, insects are defended against insecticides via multiple enzyme systems (*Terriere, 1984*; *Serebrov et al., 2006*). *Karban & Agrawal (2002)* suggested that herbivore insects are adapted to host secondary substances through physiological changes. In that case, the metabolism of plant secondary substances and chemical pesticides through enzyme systems are usually considered to be identical or similar (*Brattsten, 1988*; *Snyder & Glendining, 1996*). Various detoxification enzymes such as glutathione-S-transferase (GST) and carboxylesterase (CES) are most commonly involved in insects defense against insecticides (*Serebrov et al., 2006*). Acetylcholinesterase is a key enzyme catalyzing the hydrolysis of the neurotransmitter, acetylcholine, in the nervous system in various organisms (*Oehmichen & Besserer, 1982*; *Wang et al., 2004*). It is well-known that altered AChE is one of the main mechanisms of resistance in many insect pests that are affected by chemical and botanical insecticides (*Miao et al., 2016*). Peroxidase (POD) is an antioxidant enzyme that can provide defense against pathogens and insecticides (*Felton & Summers, 1995*; *Miao et al., 2016*). Previous studies have demonstrated that POD can be quickly up-regulated in response to xenobiotic threats and that increased activity of this enzyme is related to pesticide resistance (*Miao et al., 2016*).

Therefore, the biological traits of TS prompt us to test if: (i) TS exerts a strong lethal effect on the pest *E. obliqua*, while being non-toxic to predators such as *E. tricuspidata* and *E. albaria* in the laboratory; (ii) the multiple enzyme system in 3rd-instar *E. obliqua* is involved in defending the larvae against TS; (iii) 30% TS (w/v) shows effective control of *E. obliqua* larvae and does not harm natural enemies in tea plantations.

## MATERIAL AND METHODS

### Test insects and spiders

Larvae of *E. obliqua*, together with *E. obliqua*, spiderlings of *E. tricuspidata* and *E. albaria* were originally collected from tea bushes at the Wang Dazhen tea plantation (30.011°N, 114.363°E), Xianning, Hubei Province, China, from May to October 2014. The total area of the collection site is about 6.5 ha with parallel rows of tea plants about 100 m long and 1 m apart (Fig. S1). The site is an organic tea plantation in which no insecticides have been applied. *E. obliqua* Larvae were fed on fresh tea leaves and reared for five generations in self-made plastic chambers (10 cm diameter ×10 cm height) at 28 ± 1 °C and 75 ± 5%

relative humidity under a 14-h light:10-h dark photoperiod in the Centre for Behavioral Ecology and Evolution (College of Life Sciences, Hubei University). A chamber was used for rearing 10 larvae, and 3rd-instar larvae of *E. obliqua* were used in the following experiments.

Spiders were kept individually in glass tubes (1.5 cm diameter × 10 cm length). Each glass tube was blocked with a cotton plug and included 1 cm of moist sponge at the bottom to maintain high humidity. The tubes were kept in an illumination incubator (25 ± 1 °C, 75 ± 5% relative humidity and under a 14-h light:10-h dark photoperiod). Wild-type fruit flies (*Drosophila melanogaster*) were provided twice a week as food. Adult spiders of similar size were used for the toxicity tests.

## Reagents

Tea saponin (98% purity) was purchased from Wuhan Bai Ming Technology Co., Ltd, Hubei province, China. 10% bifenthrin (Bi) EC and 50% diafenthiuron (Di) SC were purchased from Jiangsu Dongbao Chemical Corporation Ltd., Jiangsu province, China. Both of these chemical insecticides are of low toxicity and are widely applied in the tea growing area in China to control leaf-feeding insects (*Wu et al., 2013*; *Liu, 2014*).

## Toxicity of TS to *E. obliqua* larvae

The leaf-dip bioassay method described by *Beloti et al. (2015)* and *Liang, Gao & Zheng (2003)* was adopted for the toxicity assay in which the toxicity of TS to 3rd-instar *E. obliqua* larvae was assayed. We evaluated five TS concentrations (18.75, 37.5, 75, 150 and 300 mg/mL). Distilled water was used to prepare the dilutions. Tea leaf discs (diameter 4 cm) were dipped for 20 s in one of the TS concentrations. Then, the leaf discs were dried by placing them in a glass Petri dish (diameter 9 cm). Leaf discs in control group were dipped in distilled water as described above. Thirty 3rd-instar *E. obliqua* were starved for 24 h and then transferred to the glass Petri dish (two leaves per Petri dish). Three replicates were made for each concentration. Larvae were considered to be dead if they did not respond when lightly prodded with a hair brush. Larvae mortality (%) were quantified after 48 h of treatment. Surviving larvae were, within 1 h, used for the following enzyme activity assays.

## Toxicity of insecticides to spiders

For the insecticide treatment, TS powder was diluted with distilled water to a concentration of 300 mg/mL; 10% Bi EC and 50% Di EC were diluted with distilled water to concentration of 0.01 mg/mL and 0.05 mg/mL (advised by the manufacturer of the chemicals), respectively. Prior to insecticide treatment, spiders were individually anaesthetized by carbon dioxide. The toxicity assay was conducted according to *Deng et al. (2006)*. Two droplets (0.5 µL each) of insecticide solution, by the use of a 5-µL microsyringe, were applied to the dorsal abdomen of each spider; distilled water was employed as the control. To reduce possible variation in response to treatments caused by differences in sex, spiders were randomly selected. After insecticide application, spiders were kept in Petri dishes in which one or two pieces of moist sponge were used to maintain humidity. Twenty individuals were used for each treatment and the procedure was triplicated. Spider mortality after 48 h was calculated as: (number of dead spiders/total number of spiders) ×100%.

## Assays of enzyme activity

We randomly collected three surviving 3rd-instar *E. obliqua* larvae from each TS solution and the procedure was repeated four times. Larvae were weighed and placed in a glass homogenizer in which physiological saline ($w/v = 1 : 9$) was used for homogenization. Samples were centrifuged at $10,000 \times$ g for 10 min at 4 °C. The supernatant from this final centrifugation was used to determine enzyme activities and protein concentration of each sample.

GST, CES, AChE and POD activities were determined by the use of commercial assay kits (Nanjing Jiancheng Bioengineering Institute, Nanjing, Jiangsu China) according to the manufacturer's instructions. GST, CES, AChE and POD activities were assayed in units of U/mg. Sample protein concentrations were estimated using the method described by *Bradford (1976)*. Bovine serum albumin was used for the calibration curve. Detections were performed at 595 nm emitted by a microplate reader with SoftMax Pro 6.3 software (Molecular Devices Corporation, Sunnyvale, CA, USA).

## Comparative controlling efficacy of 30% TS (w/v) and chemical insecticides against the *E. obliqua* larvae in tea plantation

To evaluate the controlling efficacy of 30% TS along with 10% Bi EC (at the recommended dose of 7.5 g a.i. ha$^{-1}$) and 50% Di EC (at the recommended dose of 45 g a.i. ha$^{-1}$) against *E. obliqua* larvae, field trials were conducted on dry days from June to July 2015 in Wang Dazhen tea plantation (30.011°N, 114.363°E) Xianning, Hubei Province, China following a randomized block design (*Roy, Mukhopadhyay & Gurusubramanian, 2010*; *Kawada et al., 2014*) with three replicates. Each identical plot (20 m$^2$) was separated by two buffer rows of non-treated tea bushes. Two rounds of foliar spray were applied by the use of a 16 L capacity knapsack sprayer equipped with a hollow cone nozzle (droplet diameter 1.2 mm, distance between nozzle and tea leaves was 30–40 cm) at 750 L ha$^{-1}$. We also established untreated control plots which involved application of clean water. A pre-treatment sampling was carried out in the respective plots on five randomly selected tea canopies. After spraying, post-treatment sampling took place at 1, 3, 5 and 7 d in each treatment plot from 4:00–5:00 pm (Beijing time) in the respective plots in the five randomly selected tea canopies. *E. obliqua* larvae were sampled from the randomly selected tea canopies, using a sweep-net (diameter 40 cm) by beating the tea canopy 10 times with a stick. To investigate the safety of 30% TS to natural enemies (spiders), the spiders were sampled before and seven days after insecticide application at five tea canopies per treatment plot. All sampled tea canopies were of similar size. Sweep-net sampling was used to sample spiders from the tea canopies. The sweep-net was held beneath the tea canopies, which was struck five times with a stick. All spiders were individually put into 1.5 mL microcentrifuge tubes containing 100% ethanol, and then kept on ice until returned to the laboratory (*Yang et al., 2017*). According to the classification of toxicity issued by the IOBC (International Organization of Biological Control), the insecticides tested under the field conditions were classified as N, harmless or slightly harmful (0–50% reduction); M, moderately harmful (51–75% reduction); or T, harmful (75% reduction) (*Boller et al., 2005*).

**Table 1  Toxicity of TS solution to 3rd-instar *Ectropis obliqua* larvae**

| Concentration of TS (mg/mL) | Mortality of larvae (% ± SE) | LC-P line | LC$_{50}$ | 95% FL (mg/mL) | $r^2$ |
|---|---|---|---|---|---|
| 300 | 66.67 ± 3.85a | $y = 4.18x{-}4.27$ | 164.32 | 126.62–233.27 | 0.898 |
| 150 | 43.33 ± 3.85b | | | | |
| 75 | 30.0 ± 1.92c | | | | |
| 37.5 | 27.78 ± 1.11c | | | | |
| 18.5 | 13.33 ± 1.93d | | | | |
| 0 | 1.11 ± 1.11e | | | | |

Notes.

TS, tea saponin; LC$_{50}$, Lethal concentration 50, the concentration causing 50% mortality;  FL,  fiducial limits (mg/mL);  SE, standard error of the means.

Mortalities (% ± SE) followed by the same letters represented no significant difference (least-significant difference test at the 5% level of significance).

Mean population reduction of pests per treatment was calculated using the following formula:

$$\text{Population reduction (PR)} = [(\text{Pre-treatment count} - \text{Post-treatment count})/$$
$$\text{Pre-treatment population count}] \times 100\%$$

$$\text{Controlling efficacy (CE)} = [(\text{PR of reagent treatment} - \text{PR of clean water treatment})/$$
$$(1 - \text{PR of clean water treatment})] \times 100\%.$$

## Statistical analysis

The LC$_{50}$ and its fiducial limits were determined by logistic regression based on the concentration probit-mortality (*Finney, 1971*). Mortality variables were expressed in percentages and the data transformed to arcsine square root. The mortality differences between larvae and adult spiders, and controlling efficacy and number of spiders were compared by using the least-significant difference (LSD) test at the 5% level of significance. The differences in enzyme activities of larvae were compared by using the unpaired Student's $t$-test at the 5% level of significance. Statistical analyses were performed using SPSS 20.0 (IBM Corp Version 20.0, IBM SPSS Statistics for Windows; IBM, Armonk, NY, USA) and Prism 5 (GraphPad Software, La Jolla, CA, USA) software.

## RESULTS

### The toxicity of TS solution to 3rd-instar *E. obliqua* larvae

Mortality of 3rd-instar *E. obliqua* larvae was in direct proportion to the five TS concentrations at 48 h with values of 13.33%, 27.78%, 30.0%, 43.33% and 66.67% (Table 1), respectively. The LC$_{50}$ value of TS solution to the 3rd-instar *E. obliqua* larvae was 164.32 mg/mL.

### Toxicity of insecticides to spiders

As no individuals died in the control test within 48 h of distilled water treatment, no adjustment for corrected mortality was necessary. The results of toxicity assay are shown in Table 2. The mortality of *E. tricuspidata* adults in the group where 300 mg/mL TS

**Table 2** Mortalities of *Ebrechtella tricuspidata* and *Evarcha albaria* adults after 48 h of treatment using different reagents.

| Treatment | Concentration (mg/mL) | Mortality (mean ± SE) (%) | |
|---|---|---|---|
| | | *E. tricuspidata* | *E. albaria* |
| 10% Bi EC | 0.01 | 80.00 ± 5.77a | 73.33 ± 3.33a |
| 50% Di SC | 1.2 | 43.33 ± 3.33b | 36.67 ± 6.67b |
| TS | 300 | 16.67 ± 3.33c | 20.00 ± 5.77c |
| Control | | 0d | 0d |

Notes.
Bi, bifenthrin EC; Di, diafenthiuron SC; TS, tea saponin; SE, standard error of the means.
Mortality (% ± SE) followed by the same letters represented no significant difference (least-significant difference test at the 5% level of significance).

solution was used was 16.67%, significantly lower than that in groups where 10% Bi EC ($F_{1,4} = 23.63$, $p < 0.01$) and 50% Di EC ($F_{1,4} = 62.74$, $p < 0.01$) were used. The mortality of *E. albaria* adults was 20%, significantly lower than that in groups where 10% Bi EC ($F_{1,4} = 23.49$, $p = 0.01$) and 50% Di EC ($F_{1,4} = 46.83$, $p < 0.01$) were applied.

## Effects of 30% TS (w/v) on enzyme activities in 3rd-instar *E. obliqua* larvae

After treatment with 30% TS, the activities of GST in 3rd-instar *E. obliqua* larvae showed a significant increase at 6 h ($t$-test, $t = 24.84$, $df = 6$, $p < 0.001$), 12 h ($t$-test, $t = 35.89$, $df = 6$, $p < 0.001$) and 24 h ($t$-test, $t = 25.01$, $df = 6$, $p < 0.001$); they, however, exhibited decrease in the later period (Fig. 1). There was no significant difference ($t$-test, $t = -2.18$, $df = 6$, $p = 0.072$) between the results with 30% TS treatment and distilled water treatment at 48 h, while the former (namely results with 30% TS) was significantly lower than the latter (namely results with distilled water) at 96 h.

The 30% TS solution significantly inhibited ($t$-test, $p < 0.001$) the activities of CES in *E. obliqua* 3rd-instar larvae (Fig. 2), and the activities of CES were maintained at a low-level over the experiment period.

As shown in Fig. 3, the activities of AChE in *E. obliqua* 3rd-instar larvae were significantly inhibited ($t$-test, $p < 0.001$) by 30% TS during the whole experimental period.

After treatment with 30% TS, the activities of POD in 3rd-instar *E. obliqua* larvae increased significantly ($t$-test, $p < 0.01$) during the whole experimental period. An exception was the 48 h when no significant difference ($t$-test, $t = 0.363$, $df = 6$, $p > 0.05$) was observed in comparison to the control (Fig. 4).

## Comparative controlling efficacy of 30% TS (w/v) and chemical insecticides against the *E. obliqua* larvae in the tea plantation

The controlling efficacies of 30% TS, 10% Bi EC and 50% Di EC against the larvae of *E. obliqua* under field conditions are shown in Table 3. Controlling efficacy (CE) was significantly lower ($F_{2,6} = 16.44$, $p < 0.01$) in plots sprayed with 30% TS than that in plots sprayed with 10% Bi EC and 50% Di EC during the first 3 d period posttreatment. Further, the CE of 30% TS was equivalent to chemical pesticides at 5 d ($F_{2,6} = 1.658$, $p > 0.05$) and 7 d ($F_{2,6} = 0.538$, $p > 0.05$), respectively.

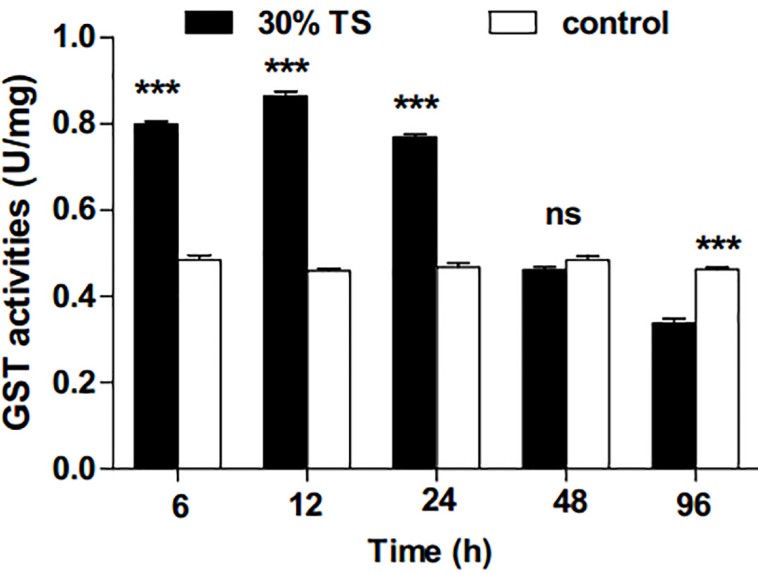

**Figure 1** **The effects of 30% (w/v) TS on GST activity in 3rd-instar larvae of *Ectropis obliqua* at different times.** TS, tea saponin; control: distilled water. Each value represents the mean of three replicates from four parallel experiments. Student's $t$-test, *, $p < 0.05$; **, $p < 0.01$; ***, $p < 0.001$; ns, no significant differences. The bars represent the standard error.

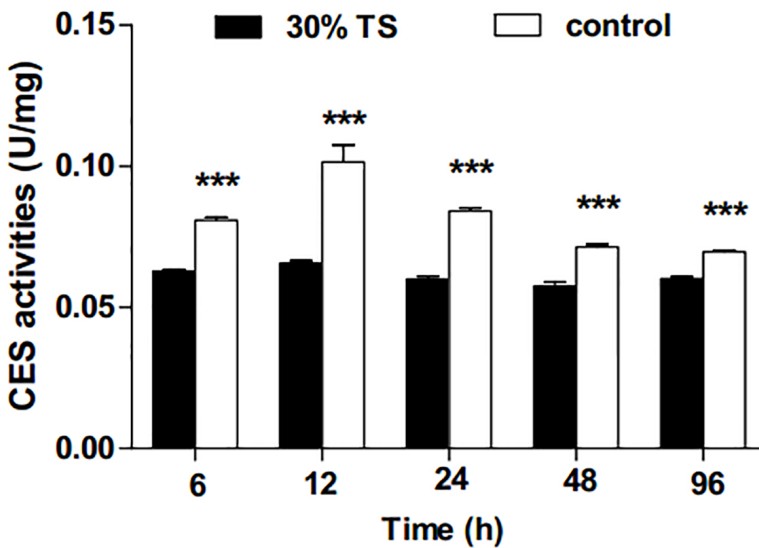

**Figure 2** **The effects of 30% (w/v) TS on CES activity in 3rd-instar larvae of *Ectropis obliqua* at different times.** TS, tea saponin; control: distilled water. Each value represents the mean of three replicates from four parallel experiments. Student's $t$-test, *, $p < 0.05$; **, $p < 0.01$; ***, $p < 0.001$; ns, no significant differences. The bars represent the standard error.

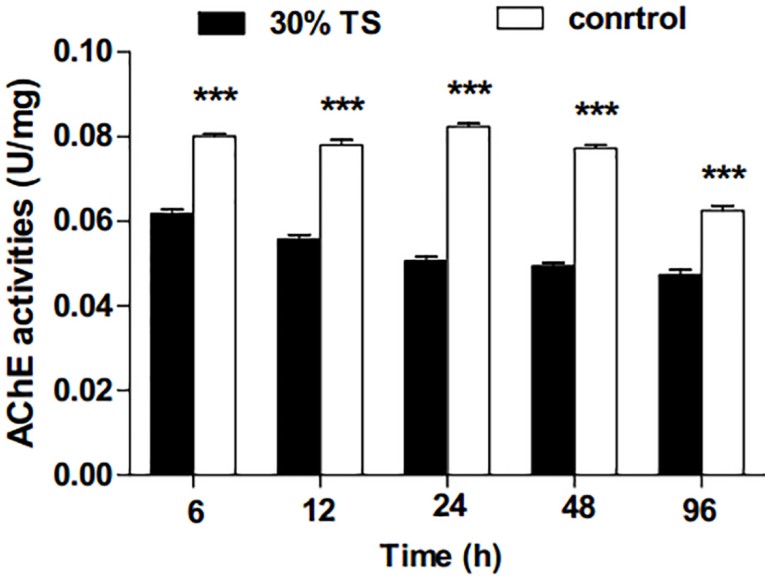

**Figure 3** **The effects of 30% (w/v) TS on AChE activity in 3rd-instar larvae of *Ectropis obliqua* at different times.** TS, tea saponin; control: distilled water. Each value represents the mean of three replicates from four parallel experiments. Student's $t$-test, *, $p < 0.05$; **, $p < 0.01$; ***, $p < 0.001$; ns, no significant differences. The bars represent the standard error.

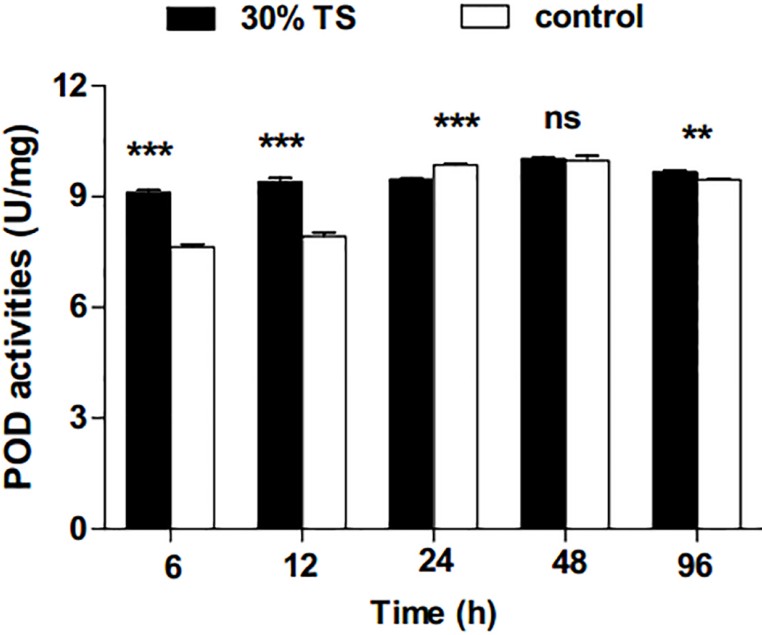

**Figure 4** **The effects of 30% (w/v) TS on POD activity in 3rd-instar larvae of *Ectropis obliqua* at different times.** TS, tea saponin; control: distilled water. Each value represents the mean of three replicates from four parallel experiments. Student's $t$-test, *, $p < 0.05$; **, $p < 0.01$; ***, $p < 0.001$; ns, no significant differences. The bars represent the standard error.

**Table 3** The controlling efficacy of 30% TS (w/v) and chemical insecticides against the *Ectropis obliqua* larvae.

| Treatment | Dose (g a.i. ha$^{-1}$) | Controlling efficacy (CE) (Mean ± SE) (%) | | | |
|---|---|---|---|---|---|
| | | 1 d | 3 d | 5 d | 7 d |
| 10% Bi EC | 7.5 | 71.23 ± 8.77a | 85.87 ± 4.07a | 60.12 ± 4.56a | 49.65 ± 3.04a |
| 50% Di SC | 45 | 56.53 ± 3.30a | 83.35 ± 4.39a | 61.32 ± 5.24a | 52.45 ± 3.72a |
| 30% TS | 562.5 | 15.93 ± 2.58b | 49.16 ± 6.40b | 77.02 ± 3.93a | 58.87 ± 4.44a |

Notes.

Bi, bifenthrin; Di, diafenthiuron; TS, tea saponin; SE, standard errors of the means.
Within columns, data (% ± SE) followed by the same letters represented no significant difference (least-significant difference test at the 5% level of significance).

**Table 4** The toxicity classes of different reagents in spiders in the treatment plots.

| Treatment | Number of spiders (mean ± SE) | | PR (%) | TC |
|---|---|---|---|---|
| | PTC | 7 d | | |
| 10% Bi EC | 6.33 ± 0.33a | 2.67 ± 0.33cd | 57.94 ± 4.82 | M |
| 50% Di SC | 5.67 ± 0.67a | 2.00 ± 0.58d | 66.27 ± 5.16 | M |
| 30% TS | 6.67 ± 0.33a | 4.67 ± 0.33b | 29.17 ± 6.25 | N |
| Control | 6.00 ± 0.57a | 6.67 ± 0.67a | −11.42 ± 5.95 | N |

Notes.

Bi, bifenthrin; Di, diafenthiuron; TS, tea saponin; Control, water spray; SE, standard errors of the means; PTC, pre-treatment count; PR, population reduction = [(PTC–7 d count)/PTC] × 100%; TC, toxicity classes (N, harmless or slightly harmful at the 0–50% level of PR; M, moderately harmful at the 51%–75% level of PR; T, harmful at over the 75% level of PR).
Within columns, the same letters represented no significant difference (least-significant difference test at the 5% level of significance).

We investigated the number of spiders in different trial plots (Table 4). The number of spiders in the plots treated by 30% TS was higher than that with 10% Bi EC ($F_{1,4} = 18.00$, $p < 0.05$) and 50% Di EC ($F_{1,4} = 16.00$, $p < 0.05$). In terms of IOBC categories for spiders, treatments of clean water and 30% TS were classified as N (harmless or slightly harmful), whereas 10% Bi EC and 50% Di EC treatments were classified as M (moderately harmful).

## DISCUSSION

Control of *E. obliqua* larvae has been mainly achieved by using synthetic chemical insecticides; however, these insecticides are extremely toxic to non-target organisms and the environment (*Potter et al., 2010*). In this study, we investigated the toxicity and controlling efficacy of 30% TS against *E. obliqua* larvae in the field in order to evaluate its use as a new and natural insecticide.

TS has been used for managing pests because of its insecticidal properties, which include stomach poisoning and odor repellence (*Yang & Zhang, 2012*; *Cai et al., 2016*). In our study, 30% TS showed insecticidal activities, causing dose-dependent mortality (66.67%) in 3rd-instar *E. obliqua* larvae. Our results were in accordance with *Chen et al. (1996)*, who demonstrated that 25% active ingredient of TS-D solution significantly increased larval mortality in the cabbage butterfly (*Pieris rapae*). A similar result was demonstrated

by *Rizwan-ul Haq et al. (2009)*, who reported that TS showed increased efficiency against *Spodoptera* when combined with *Bacillus thuringiensis* Kurstaki. In addition, *Bandeira et al. (2013)* reported that ethanolic extracts of the flowers and fruits of *Muntingia calabura* were toxic to diamondback moth (Plutella xylostella) larvae, which can partly support our findings.

The beneficials *E. tricuspidata* and *E. albaria* are predators of the larvae of *E. obliqua,* and thereby important non-target species in tea plantation can easily be affected by insecticides. Susceptibilities of these species to the 30% TS, 10% Bi EC and 50% Di EC were assessed in this study. After 48 h, the mortalities of *E. tricuspidata* and *E. albaria* which were treated with 30% TS were significantly higher than those in groups where clean water was applied, while being significantly lower than the mortalities of the two species treated with 10% Bi EC and 50% Di EC (Table 2). These results indicated that in the laboratory 30% TS which exerted lethal effect on the two spider species was less harmful than the two chemical insecticides. It is clear that acute toxicity tests do not reflect the full range of effects of a compound on an organism. Our study mainly focused on acute toxicity tests which can reveal details of intoxication but often underestimate mortality in comparison with field studies, which only take into account one route of uptake (*Wiles & Jepson, 1992*; *Pekár, 2012*). The effects that an insecticide has on a beneficial species in the field are complex processes involving both susceptibility and exposure, with exposure being a multidimensional process (*John, Paul & Daniel, 1995*). *Xavier et al. (2015)* demonstrated that the acute toxicity of botanical insecticides might involve delayed effects. In addition, insecticides affect virtually all life-history trails of spiders (*Pekár, 2012*), whereas the long-term effects of TS on these spiders are currently unknown.

Our findings also indicated that 30% TS exerted remarkable effects on the activities of detoxification enzymes. Insect resistance is determined by the activities of detoxifying enzymes and decreased target sensitivity to chemical pesticides (*Felton & Summers, 1995*; *Potter et al., 2010*). The changes usually involve increased detoxification enzyme activities and introduction of additional isoforms (*Miao et al., 2016*). Increased activity of detoxifying enzymes in insects represents a response to intoxication with insecticides or xenobiotics (*Singh & Singh, 2000*; *Serebrov et al., 2006*; *Gopalakrishnan et al., 2011*). *Rizwan-ul Haq et al. (2009)* evaluated the bioactivities of TS solution in *Spodoptera exigua* (Lepidoptera: Noctuidae), which provides some helpful information about activities of enzymes against TS which is involved in the resistance mechanism in insects. In this study, we found that the activities of GST significantly increased during the initial period following TS treatment, which suggests that this enzyme may act to detoxify TS. Meanwhile, the activities of CES decreased significantly. In the case of CES, there are two mechanisms of resistance. One is that the over-expressed esterase proteins work like "sponges" to sequester insecticide molecules rather than hydrolyzing them (*Chevillon et al., 1999*). The other mode of resistance results from changes in the enzymatic properties of CES, specifically, increased activity towards insecticides and decreased activity towards generic substrates (*Feng et al., 2015*). Therefore, the inhibition of CES in our study may result in obstruction of CES to degrade or sequester TS. We speculated that TS may increase susceptibility of *E. obliqua* larvae to the insecticide by inhibiting CES activities. AChE is a key enzyme in the nervous

system of various organisms, which can terminate nerve impulses by catalyzing the hydrolysis of the neurotransmitter acetylcholine (*Wang et al., 2004*; *Senthil et al., 2008*). It is well known that altered AChE is one of the main mechanisms of resistance in many insect pests (*Serebrov et al., 2006*). Our study showed that AChE activities decreased significantly. We inferred that 30% TS inhibited the AChE activities, thus causing accumulation of ACh at the synapses. That explains the reason why the post-synaptic membrane is in a state of permanent stimulation. This result in insect paralysis, ataxia, general lack of co-ordination in the neuromuscular system, and eventual death (*Singh & Singh, 2000*). POD is the key antioxidant enzyme that can be quickly up-regulated in response to natural penetrating xenobiotics (*Wu et al., 2011*), and the increase of POD activities is related to pesticide resistance and melanization in insects (*Terriere, 1984*; *Potter et al., 2010*). It is shown that POD activities increased over the whole experimental period, except 48 h. We assumed that the enhanced activities of POD are associated with the elimination of ROS. Large quantities of generated ROS can rapidly denature a wide range of biomolecules, thereby threatening virtually all cellular processes and leading to insect death (*Felton & Summers, 1995*).

We found that the enzymatic defense against TS assault in 3rd-instar *E. obliqua* larvae was generally activated. Our results reveal that the perturbation of enzyme activities by TS seems to be one of the underlying modes of action. In addition, several studies documented another probable mode of action of saponins involving interaction with membrane cholesterol, which causes membrane destabilization and provokes cell death (*Sung, Kendall & Rao, 1995*; *Hu, Konoki & Tachibana, 1996*; *Chaieb et al., 2007*). This interaction structurally modified the phospholipid double layer which would be at the origin of disturbances of cellular exchanges leading to cytotoxicity. *Chaieb et al. (2007)* by the use of histological methods, demonstrated a cytotoxic effect of crude saponic extract on the fat body of *Spodoptera* larvae, and cell destruction of the foregut and gastric caeca of *Schistocerca*. The same results were reported by *Gögelein & Hüby (1984)* and *Hu, Konoki & Tachibana (1996)*. Therefore, TS exerted multiple modes of action involving enzymatic and physiological perturbation on the 3rd-instar *E. obliqua* larvae.

The effectiveness of 30% TS and two types of chemical insecticides against *E. obliqua* larvae in the field were investigated in this study. As a botanical production, the controlling efficacy of 30% TS was exceeded by 10% Bi EC, 50% Di EC at 5 d and 7 d, although the difference was not significant (Table 2). A previous study proposed that natural enemies should be the first consideration in any pest management intervention (*Koul & Dhaliwal, 2003*). Any integrated approach to pest management must be compatible with natural enemy conservation (*Amoabeng et al., 2013*). *Yang et al. (2017)* employed comprehensive indices for evaluating the predation of *E. obliqua* by nine common spider species in Chinese tea plantations. Although the pooled average number of spiders with TS treatment was significantly lower than that with clean water application (Table 4) after 7 d of reagent application in this study, we supposed that this lower abundance of spiders in experimental plots may be due to reduced prey availability (*Sunderland, 1992*; *Markó et al., 2009*). Because TS exerts strong fungicidal activity, it can reduce the abundance of fungi. The reduction in levels of fungi may reduce the abundance of mycetophagous pests such as springtails and some beetles, thus changing the prey availability for spiders

(*Sunderland, 1992*). Direct evidence of this has not been reported so far. However, pyrazophos applied in the field decreased spider abundance (*Volkmar & Wetzel, 1993*), although laboratory tests showed that this fungicide is harmless to spiders (*Mansour, Heimbach & Wehling, 1992*). *Peng, Peng & Zeng (2017)* have reported that 30% TS exerted a significantly lower repellent rate to spiders compared with chemical insecticides, which could partly support our results. Our results indicated that 30% TS should be placed in the class N (harmless or slightly harmful) of IOBC categories for natural enemies, namely spiders. Therefore, the distribution of spiders can indicate that TS is relatively less harmless than chemical insecticides. In addition, the procedure for preparation of 30% TS is simple, which only involves the use of water, TS production is cheap and TS is readily available. Our findings suggest a perspective for controlling *E. obliqua* larvae without the use of chemical insecticides. Using TS to control *E. obliqua* larvae would help the tea industry in many ways, such as freeing tea products of residues, reducing pesticide load, as well as guaranteeing cost effectiveness and customer satisfaction (*Roy, Mukhopadhyay & Gurusubramanian, 2010*).

## CONCLUSION

In conclusion, our results indicated that 30% TS, as a new alternative biocontrol insecticide, has significant potential against the *E. obliqua* larvae. It involves in multiple modes of action and exerts only slightly harmful effects on the natural enemies such as spiders in field applications. As a natural product that is abundant in tea plantations, 30% TS could be effectively utilized in the IPM envisaged for tea.

## ACKNOWLEDGEMENTS

We are grateful to the Wang Dazhen tea plantation for giving permission to conduct the field research in Xianning, China.

### Funding

This work was supported by State's Key Project of Research and Development Plan, No. 2016YFD0200900 (Peng Yu), the National Natural Science Foundation of China, No. 31672317 (Peng Yu), and the Competitive Planning Projects of Hubei Academy of Agricultural Sciences, No. 2016jzxjh012 (Peng Yu). There was no additional external funding received for this study. The funders had no role in study design, data collection and analysis, decision to publish, or preparation of the manuscript.

### Grant Disclosures

The following grant information was disclosed by the authors:
State's Key Project of Research and Development Plan: 2016YFD0200900.
National Natural Science Foundation of China: 31672317.
Hubei Academy of Agricultural Sciences: 2016jzxjh012.

## Competing Interests

The authors declare there are no competing interests.

## Author Contributions

- Chi Zeng and Yu Peng conceived and designed the experiments, performed the experiments, analyzed the data, contributed reagents/materials/analysis tools, prepared figures and/or tables, authored or reviewed drafts of the paper, approved the final draft.
- Lingbing Wu analyzed the data, prepared figures and/or tables, authored or reviewed drafts of the paper, approved the final draft.
- Yao Zhao and Yueli Yun contributed reagents/materials/analysis tools, prepared figures and/or tables, authored or reviewed drafts of the paper, approved the final draft.

## Data Availability

The raw data are provided as a Supplemental Information 1.

## Supplemental Information

Supplemental information for this article can be found online at http://dx.doi.org/10.7717/peerj.4534#supplemental-information.

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
