# Peer review of "Tea saponin reduces the damage of Ectropis obliqua to tea crops, and exerts reduced effects on the spiders Ebrechtella tricuspidata and Evarcha albaria compared to chemical insecticides"

_PeerJ, doi:10.7717/peerj.4534_

## Round 0.1 · original submission · Major Revisions

The study is an assessment of the impact of a botanical insecticide on an important pest and associated beneficial insects in the lab and the field. It is relevant from both a crop protection and an environmental perspective. Overall, the study has been thoroughly conducted and the methods are appropriate. There are a number of points that the reviewers and myself have found to need improvement.

The English must be revised in the whole manuscript. It is imperative that the authors take care of this before the manuscript can be accepted.

The article is currently lacking some background on the natural enemies of the pest, and whether the two chosen spider species are important for the natural control of the pest. This is key to the framing of the research question.

The reviewers highlight that some information is lacking before one can consider the study as being reproducible.

While adequate controls were included, the controls are not always reported adequately (see comments by reviewer #2).
.
The data on which the conclusions are based must be provided or made available in an acceptable discipline-specific repository. This is not currently the case. Data in tables and figures are secondary data, not primary data.

As for the conclusions, I would like to point out that it is increasingly clear that acute toxicity tests do not reflect the full range of effects of a compound on an organism. The article should include as discussion of the limitations of the approaches used in assessing non-target effects.

Please review the formatting of the reference list and make sure it adheres to the PeerJ format. Currently, this is not the case, for instance, the issue is listed in addition to the volume.

Please take care to carefully respond to the reviewer comments, which contain further valuable points.

·

Basic reporting

The indoor and outdoor toxicities of tea saponin on insect pest Ectropis obliqua and two spider species, Ebrechtella tricuspidata and Evarcha albaria, were tested. The effect of tea saponin on some enzymes in Ectropis obliqua was also investigated.The results will help to find more eco-friendly insecticides that can be used in the tea garden.
However, there are many spelling or even grammar mistakes in the paper. I have colored some. I do suggest the authors to find some English native speakers to help to improve the writing.

Experimental design

Generally speaking, the experiments were well designed. But the method still needs to be introduced in more detail.

Validity of the findings

The data were well analysed and the conclusions are well stated.

Reviewer 2 ·

Basic reporting

The article feet generally standards, but some literature references (given below) can be added to ameliorate discussion of the results.
some lack in introducing the research:
Abstract
the background of the article only talks about the effect of TS on Ectropis obliqua, while all the article talks about the caterpillar and two other spiders. Try to include work on spiders at the abstract level.

introduction
The author does not say why he chose these two spiders to do his study, are they predators of Ectropis?, it must be specified and prove it using references on relation between Ectropis and spiders.

Experimental design

some lacks in describing material and methods:

in Toxicity assay of insecticides to spiders
- This part to put right after the tests on the larvae of E.obliqua and before the enzymatic activity

- A dilution of 0.01 and 0.05 was made for both chemicals, explained why you used these dilutions, are these dilutions advised by the manufacturer of chemicals?.

- Say whether the spiders undergo continuous breeding or if they are simply kept up to experience. In the second case, say whether the author has verified that the individuals coming from the field are not previously treated with insecticides ...

- The number of spiders and the number of repetitions used in the test was not specified. Please give these informations in material and method

Validity of the findings

this article gives some important results, but these results must be more discussed with articles dealing with mode of action of saponins.

- Line 245 pea aphid is not Spodoptera littoralis, and cotton leafworm caterpillars is not (Acyrthosiphon pisum). There is inversion in scientific names.
- Discussion on the physiological effect of saponin is done without discussing one of the most documented mode of action: interaction with cholesterol, and cytotoxicity
Try to discuss this with some references as:
http://www.iresa.tn/tjpp/tjpp9/tjpp9/4Ikbal.pdf
http://www.scialert.net/qredirect.php?doi=jbs.2007.95.101&linkid=pdf
https://www.researchgate.net/publication/6421088_Insect_growth_regulator_activity_of_Cestrum_parqui_saponins_an_interaction_with_cholesterol_metabolism

Additional comments

- title must be changed
The author says in the title that saponins do not cause significant harm on the tested spiders. Significant habitually compared to control? in Table 2 there is spider mortality of 16.6 and 20%, but here you do not report the mortalities of control.

So to be able to say that the mortalities are not significant, it is necessary to put the values of mortality of the non treated control (even if the latter are null) since here it is an uncorrected mortality, and also it is necessary to make the comparison of the average between the control and the TS to be able to say that the mortalities are not significant.

In addition to Table 4, which relates to the garden test, there is a significant difference between control and 30% TS, so there is an effect of saponins on spiders.
I propose to change the title, replace "non significant harm" by "reduiced effect"

Reviewer 3 ·

Basic reporting

Title: Specify the TS species and the evaluated spiders.

You have to be careful in some terms and spaces when writing the text. Line 31, indoor, suggestion, laboratory. Line 41, were inhibited.

Summary: Background, does not talk about spiders and their importance in the work. Methods, does not specify the use of insecticide products against spiders and is an issue that is addressed in the results section.

Introduction: line 56: Camellia sinensis is tea saponin or is the extract of the plant which is called like that? The first paragraph can be linked to the paragraph on line 85-91 since the information is related.

Line 67: E. oblicua, attacks all types of tea crops or only some species? Be more specific.

Figures: They look with a lot of resolution. See if it is the right amount of pixels. The tables are well structured. Review writing in figure 3.

hyphotheses
It is necessary to specify why the evaluation of the enzymes is important for this work in the introduction, on the basis of which, to formulate the hypothesis again. Be clearer previously in the introduction of why use 30% TS in the field, since it is assumed that this is the dose or quantity that will have an effect on the larvae. Eliminate We hope to gain useful information to extend the application of TS, this phrase is already a perspective of the work.

Experimental design

The study gives us the idea of having more extracts or compounds of plants with insecticidal activity to avoid the use of chemical insecticides. However, it is not clear why the study of spiders, it must be more forceful in the main idea.

In some experiments in the technical standard, the analysis criteria are not defined. TS: they do not mention because they use a commercial, line 120 and not one of the tea plants of any of the bushes they mention. Larvae, specify because it was the best technique to provide ts and not formulate a diet where it is ensured that the larva consume all the concentration of TS. enzymes, they do not mention how much the units are equivalent, they leave it to the criterion of the use of the kits.

The method for the enzymatic determination is not clear, it is not specified what part of the larva were made, nor the conditions to be followed for the analysis of them, so the repetition of the experiment may be unsuccessful.

It is not clear why they decided to use Bi and Di at different concentrations, when you normally compare with the same concentration, including 30% of TS. Be clearer in these analyzes since they are the main structure of the article.

It does not make clear how the determination was made to continue the evaluation with 30% of TS. Where did the specific value come from?

Validity of the findings

The experiments are based on bibliography, but they are not explicit and therefore some results are not clear.

In Figures 1-4. The data obtained with the control show activity of the specific enzymes with water as a control. It is not explicit how this phenomenon occurs, and in their case, they do not state how good the result of the effect of adding 30% TS is. Specify the importance of the data based on the literature already reported.

The statistics in Table 3 are not clear, low, intermediate and high values share the same letter, so they belong to the same group of statistical analysis, which, it can not be possible. Review the statistics and specify the explanation in the caption or in the results section.

The conclusion agrees with what was established with the study questions and the results obtained.

Additional comments

The information generated by the authors and reflected in the article is very interesting, from the level of the generation of organic insecticides. However, it is not emphasized in providing the specific information in the methodology and in the discussion of them. The article needs to be modified in the framed sections for its probable publication.

---

## Round 0.2 · Minor Revisions

(a) The manuscript has been nicely revised and most of the points have been addressed - with some exceptions that still need to be addressed. One remaining issue is language. One of the reviewers has provided an annotated document highlighting the parts of the manuscript that need language changes, and I detail in the following a long list of other changes that I recommend. A second issue is the lack of detail with respect to the data collection in the field experiment.

L184-185 Please describe the method used to record the spiders. also: How large were the bushes? Can a photograph of the field setup be included? The spiders counts seem to have very low variance (5-7 spiders per plot before treatment), is it possible to give an explanation why this may be so?

L46 I suggest to delete ”a type of voracious worm”, which is very colloquial. The rest of the sentence makes it clear that it is a severe pest.
L50 “indiscriminate uses” by “indiscriminate use”
L55 replace “exert” by “have”
L58 replace “have multiple bioactivities” by “, as well as multiple bioactivities.”
L61 replace “pest management” by “managing pests”
L68-69 delete “their properties, causing”
L70 delete “the” in “TS has the potential”
L72 delete “and presents no harm to the environment”. This is not supported by any literature reference. On the contrary, strong insecticidal activity suggest that there is potential for harm to the environment via effects on non-target organisms.
L73 30% (w/v) TS needs to be put in words first.
L75 Delete “To date,”
L80 Do not start a sentence with “And”. Here, I suggest to just delet and start with “Yang et al. (2017)…”
L84 greater acceptability than what?
L86-87 Better: “Likewise, insects are defended against insecticides via…”
L89 it is not clear exactly what is mean with “identical and similar”. In what respect?
L103 replace “but is not toxic to predators, such as” by “whil. Ebeing non-toxic to predators such as”, and remove the comma after albaria.
L104 Instead of “participates(…) TS assault;”, write “is involved in defending the larvae against TS;”
L105 Replace “and dose(…) plantation” by “does not harm natural enemies in tea plantations”
L244 “In terms of IOBC categories for spiders, treatments of clean water were classified as X, wheras xxxxx were classified as X”
L263 Replace “As predators of the larvae of E. obliqra, E. tricuspidata and E. albaria are easily affected by insecticides and are the important non-target and beneficial species in tea plantation.” By “The beneficials E. tricuspidata and E. albaria are predators of the larvae of E. obliqra, and thereby important non-target species in tea plantation that can easily be affected by insecticides.”
L267 These results indicated that in the laboratory 30% TS had a reduced effect on the two spiders when compared to clean water.
L301 “shown” repeated twice.
L307 move the word “underlying” so that the sentence ends with “seems to be one of the underlying modes of action.”
L314 Capitalize spodoptera and schistocerca
L328 “reduced prey availability”
L339 replace “friendlier” by “less harmless”
L349 “such as spiders”

Raw data: for the field experiment the authors should include the actual raw data at the smallest unit of observation (per bush) if available, as well as the taxonomic information on the spiders collected.

(b) the manuscript cannot be accepted before the language is satisfactory, so I recommend the authors not only act upon the reviewers suggestion and the changes suggested above, but also check through the rest of the manuscript.

(c) on this basis I recommend minor revisions.

·

Basic reporting

The authors have tried their best to improve the writing. However there still many gramatical mistakes. For example, it should be written as "the toxicity to" or "on" but it was written as "in" in some places in the paper. For the insecticide name, it should be written as "the percentage of the active incgredient"+ "active ingrediant name"+"formulation name",. In the field experiment, "30% tea saponin wettable granule "was used, if I am right, however, where was this product from was not introduced. In the reference part, there are some mistakes, too.

Experimental design

Generally, the experiments were well designed. However, the counting method for the spiders was not illustrated. For enzyme activity assay, it was said that the surviving insects after insecticide treatment were used, but when were the insects were sampled?

Validity of the findings

When the authors explained the inhibition effects of tea saponin on the carboxylesterase, they said that helped to increase the susceptibility of the insects to that. However, that was not complete, because the underlying mechanisms related to the effects of carboxylesterase on insecticide susceptibility lie in two sides, one is to degrade the insecticides and the other is to seperate the insecticides from the hymolimph

Reviewer 2 ·

Basic reporting

all corrections submitted to the author were considered

Experimental design

all corrections submitted to the author were considered

Validity of the findings

all corrections submitted to the author were considered

Additional comments

all corrections submitted to the author were considered

Reviewer 3 ·

Basic reporting

Introduction.
Line 73: Add reference of the antecedent.
Line 80: Delete and before the reference of Yang et al.

Materials and methods.
Line 109: Modify the wording. Be aware that they are the larvae of the two spiders. I understand that the work is only used on adult spiders.
Lines 145 and 153: How is the percentage of mortality quantified?
Line 162: Difference of type of letter or points of the same. Make change.
Line 169: what does it mean 30% of TS (300 mg / ml)? Since it is not understood before in the text how or from where it is obtained.

Discussion.
Line 254: Mention that the results are similar, refer to other jobs reported with TS or that are similar because extracts from other plants, kill the same plague evaluated in this work? review the wording.
Line 336: Review Mansour reference is not found in the bibliography.

References.
Lines 434-436, 532-534: Differentiate the references with letter or number to locate them in the part corresponding to the article.
Line 467: The reference in the text was not found.

Experimental design

Experimental designs according to what was evaluated.

Validity of the findings

Acceptable results, conclusion according to the results and robust statistical data.

Additional comments

I know that if they corrected the article as requested. Review the details that are requested of the writing and lack of some references.

---

## Round 0.3 · Minor Revisions

I'd suggest the authors consider making their title more informative by adding a few words: "Tea saponin reduces the damage of Ectropis obliqua to tea crops, and exerts reduced effects on the spiders Ebrechtella tricuspidata and Evarcha albaria compared to chemical insecticides"

Otherwise, I still find some issues in the language, and one issue of misinterpretation of the findings. Most of these have been introduced in the revision, and a few are issues I missed at the previous read (but note that it is the authors who are to take care of language issues, not the editor)

L99 trails should be “trials”
L101 has should be have
L104 find alternative
L127 a strong lethal effect on the pest E.obliqua, while
L169 larval mortality
L170 unclear what this means “Surviving larvae were used for the following enzyme activity assays in 1 h.”
L184 Spider mortality after 48 h was calculated as: (number of dead spiders / total number of spiders) × 100%.
L206 “, 50% Di EC” by “and 50% Di EC”
L210 delete “and” in “with and three”
L214 correct the sentence: “An untreated control plot which involving application of clean water was simultaneously proceed during the study”. Example: “We also established untreated control plots which involved application of clean water.”
L221 it should be “the spiders were sampled”
L225 canopy should be singular.
L308 not sure “antibiotic” is the right word.
L320 oliqra should be obliqua
L326 this statement is a misrepresentation of the results: “These results indicated that in the laboratory 30% TS had a reduced effect on the two spiders when compared with clean water”
Indeed, it seems from Table 2 that there is no mortality in the control treatments, but 16-20% mortality in the TS treatment. These differences are statistically significant.
365-366 change to “This results in insect paralysis,”
L415 I think the sentences are a bit strange and difficult to understand. Replace “This finding could be a point of view of controlling E. obliqua larvae without the use of chemical insecticides. This approach remains important in controlling the E. obliqua larvae and would” by “Our findings suggests a perspective for controlling E. obliqua larvae without the use of chemical insecticides. Using TS to control E. obliqua larvae would…”

---

## Round 0.4 · accepted · Accept

Final copyedit: L382: "guaranting" should be "guaranteeing"